# Comparative Genomics Reveals Genetic Diversity and Variation in Metabolic Traits in *Fructilactobacillus sanfranciscensis* Strains

**DOI:** 10.3390/microorganisms12050845

**Published:** 2024-04-23

**Authors:** Xiaxia He, Yujuan Yu, Rober Kemperman, Luciana Jimenez, Faizan Ahmed Sadiq, Guohua Zhang

**Affiliations:** 1School of Life Science, Shanxi University, Taiyuan 030006, China; hexiaxia1122@163.com (X.H.); juan2035@163.com (Y.Y.); 2Lesaffre Insituut of Science and Technology, 101 Rue de Menin, 59700 Marc-en-Baroeul, France; r.kemperman@lesaffre.com (R.K.); l.jimenez@lesaffre.com (L.J.); 3Advanced Therapies Group, School of Dentistry, Cardiff University, Cardiff CF14 4XY, UK; faizan_nri@yahoo.co.uk; 4Flanders Research Institute for Agriculture, Fisheries and Food (ILVO), Technology and Food Science Unit, Burgemeester Van Gansberghelaan 92/1, 9820 Merelbeke, Belgium

**Keywords:** *Fructilactobacillus sanfranciscensis*, comparative genomics, carbohydrates utilization, exopolysaccharides, CRISPR-Cas

## Abstract

*Fructilactobacillus sanfranciscensis* is a significant and dominant bacterial species of sourdough microbiota from ecological and functional perspectives. Despite the remarkable prevalence of different strains of this species in sourdoughs worldwide, the drivers behind the genetic diversity of this species needed to be clarified. In this research, 14 *F. sanfranciscensis* strains were isolated from sourdough samples to evaluate the genetic diversity and variation in metabolic traits. These 14 and 31 other strains (obtained from the NCBI database) genomes were compared. The values for genome size and GC content, on average, turned out to 1.31 Mbp and 34.25%, respectively. In 45 *F. sanfranciscensis* strains, there were 162 core genes and 0 to 51 unique genes present in each strain. The primary functions of core genes were related to nucleotide, lipid transport, and amino acid, as well as carbohydrate metabolism. The size of core genes accounted for 41.18% of the pan-genome size in 14 *F. sanfranciscensis* strains, i.e., 0.70 Mbp of 1.70 Mbp. There were genetic variations among the 14 strains involved in carbohydrate utilization and antibiotic resistance. Moreover, exopolysaccharides biosynthesis-related genes were annotated, including *epsABD*, *wxz*, *wzy*. The Type IIA & IE CRISPR-Cas systems, pediocin PA-1 and Lacticin_3147_A1 bacteriocins operons were also discovered in *F. sanfranciscensis*. These findings can help to select desirable *F. sanfranciscensis* strains to develop standardized starter culture for sourdough fermentation, and expect to provide traditional fermented pasta with a higher quality and nutritional value for the consumers.

## 1. Introduction

Wheat- and barley-based breads have unique significance in the human diet globally as they are important dietary sources of energy, carbohydrates, and plant proteins [1]. The last decade has witnessed a rapid expansion in China’s bakery sector. The total annual sale of bread from 2015 to 2020, especially packaged and unpackaged leavened bread, reached more than $6 billion, which accounted for 18% of the total sales of China’s bakery industry [2]. Steamed bread is the conventional staple food in China [3], which undergoes fermentation by a traditional starter culture termed sourdough. Archaeological evidence suggests that baking leavened bread using sourdough originated in ancient Egypt 4000–5000 years ago [4,5]. Nowadays, sourdough is widely used in industrial and artisan bakeries in various products ranging from a loaf of bread to crackers, baguettes, pancakes, and pizzas. Sourdoughs are the product of natural or starter culture initiation, which is mainly composed of a mixture of cereal flour and water [6,7], consisting mainly of a microbial ecosystem of lactic acid bacteria (LAB), such as *Fructilactobacillus sanfranciscensis* (formerly *Lactobacillus sanfranciscensis*), *Lactiplantibacillus plantarum*, and *Pediococcus pentosaceus*, and yeast, such as *Saccharomyces cerevisiae* and *Candida humilis* [3,8,9]. Based on the texture, flavor, and nutritional final characteristics of inoculum and baked goods, four distinct types of sourdough can be distinguished: I, II, III, and IV [10].

*Fructilactobacillus sanfranciscensis* is the most prevalent and functionally important *Lactobacillus* species in the Type I sourdough. As the native key bacterium of the sourdough ecology [3,11,12], *F. sanfranciscensis* has always been of great interest to ecologists and microbiologists. *F. sanfranciscensis* can metabolize carbohydrates (e.g., sucrose or maltose) to produce exopolysaccharides (EPS) and hydrolyze proteins to peptides. These metabolites influence the flavor, nutritional quality, and shelf life of sourdough [3,13]. Interestingly, multiple studies have shown that a large number of *F. sanfranciscensis* strains are prevalent in different types of sourdoughs from diverse origins [3,14,15,16]. These strains have been shown to interact differentially with prevalent yeast species and other LAB in sourdough [8,17,18].

Researchers have demonstrated that *F. sanfranciscensis* can be isolated from sourdough and also from the gastrointestinal tract of *Drosophila melanogaster*, and tea floral pollen [3,19]. Different strains of *F. sanfranciscensis* have shown various biological and chemical functions of interest to the bakery industry. Corsetti et al. (1996) [20] reported that bacteriocins produced by *F. sanfranciscensis* C57 displayed effective antibacterial properties. Zhang et al. (2020) [21] studied the physiological and biochemical characteristics of *F. sanfranciscensis* Ls-1001, which had a strong inhibitory effect against *Staphylococcus aureus*, *Escherichia coli*, and other foodborne pathogens. Furthermore, it was discovered that the EPS produced by strain Ls-1001 had vigorous antioxidant activity in vitro [22]. Glutathione reductase activity (thiol-exchange reactions) of *F. sanfranciscensis* has been shown to reduce wheat germ agglutinin, which is linked to nonCeliac wheat sensitivity [23]. Due to its gastrointestinal resistance at lower pH, GRAS (generally recognized as safe), has proven health benefits, so there is an increasing interest in using *F. sanfranciscensis* as a probiotic in various foods [24]. To exploit the biological and physiological functions of *F. sanfranciscensis*, a deeper understanding of genetic variations in this species and the factors governing these variations is mandatory.

A genotype-phenotype study of *F. sanfranciscensis* has shown several differences in carbohydrate and external electron receptor utilization among different strains of this species [11,17,25]. Rogalski et al. (2020c) [16] also provided powerful evidence for this with the help of comparative genomics supported by physiological data. Baek et al. (2021) [26] proposed that desirable strains of this species should be selected for a standard starter culture based on their phenotypic and physiological characteristics and their ability to interact with other sourdough microbiota. However, selecting desirable strains without the knowledge of genetic drivers remains elusive. Thus, it is important to compare whole genomes of various strains of this species to determine different factors that distinguish strains and regulate the physiology, metabolism, and diversity among *F. sanfranciscensis* strains. This research provides a solid theoretical basis for selecting desirable strains of *F. sanfranciscensis* with more significant functional characteristics to enhance fermented baked goods at domestic and commercial levels.

## 2. Materials and Methods

### 2.1. Strains, Genome Sequencing and Assembly

Genomes of 45 strains of *F. sanfranciscensis* were analyzed and compared. Among the 45 *F. sanfranciscensis* strains, 14 strains were stored at our lab (Laboratory for Exploration and Utilization of Food Microbial Resources, Shanxi University, Taiyuan, China), while 31 strains genomes were obtained from NCBI. All 14 *F. sanfranciscensis* were stored at −80 °C and preserved in 30% glycerol. Strains were activated in a modified MRS (mMRS) medium [22] 2 times and then cultured at 30 °C under anaerobic conditions. All 14 *F. sanfranciscensis* strains were sequenced using the Illumina Hiseq × 10 platform (Majorbio BioTech Co., Shanghai, China) using the pair-end library of 400 bp fragments, generating 2 × 150 bp. SOAPdenovo 2.04 software was used to handle the raw sequencing reads for quality assessment [27] and assembly. The assembled fragments were constructed after filling the assembly in undetermined regions using the software of GapCloser 1.12 [27], generating the draft genomes used in subsequent analyses.

### 2.2. Genomic Characteristics Prediction and Pan-Genome and Core Genes Analysis

The GC content of 45 *F. sanfranciscensis* strains were determined. Glimmer 3.02 and Prodigal 2.6.3 software were used for coding sequence (CDS) prediction. The sequences were annotated using the Swiss-prot database (accessed on 1 November 2023) and the RefSeq nonredundant protein (NR) database (accessed on 10 November 2023) [28] with the E value set at 1 × 10^−5^.

PGAP 1.2.1 software was utilized to analyze the pan-genome size and the clustering of the functional genes [29] using all 45 strains of *F. sanfranciscensis*. Protein sequences of 45 strains of *F. sanfranciscensis* were compared using Orthomcl v2.0.9 software to create Venn diagrams based on the obtained data [30]. The clusters of orthologous groups of proteins (COGs) database (accessed on 10 November 2023) was used to classify the proteins encoded genomes.

### 2.3. Phylogenetic Analysis and Determination of the Average Nucleotide Identity (ANI) Value

The ANI values were calculated using pyani software (accessed on 15 November 2023) based on an average of the comparison of the coding sequences of all lineal homologous proteins between the two genomes, which reflected the evolutionary distance between the genomes. Then these captured data were visualized using heat map tools (https://cloud.majorbio.com/page/tools/, accessed on 18 November 2023). OrthoMCL 2.0.9 software was used to compare and summarize all the homologous genes of 45 strains of *F. sanfranciscensis* [31,32]. To further investigate the potential evolutionary relationship between these strains, MEGA 7.0 software was used to create phylogenetic trees [33].

### 2.4. Genotypic and Phenotypic Analysis of Carbohydrate Metabolism

The carbohydrate-active enzymes of all the 14 *F. sanfranciscensis* strains were categorized and analyzed using the carbohydrate active enzymes database (CAZymes, http://www.cazy.org/, accessed on 22 November 2023), using Hummer and Dianond software. The heat map in the Origin Pro software was used to draw the visual clustering relationship map.

D-glucose, D-fructose, D-trehalose, D-sucrose, D-galactose, L-arabinose provided by Shanghai McLean Biochemical Technology Co., Ltd. (Shanghai, China), and D-lactose, soluble starch, D-maltose, and mannitol provided by Beijing Aobo Star Biotechnology Co., Ltd. (Beijing, China) were used as sole carbon sources to verify the expression of carbohydrate utilization genes in 14 *F. sanfranciscensis* strains. Ten different carbohydrates were added individually into the mMRS medium at the concentration of 2 g/100 mL followed by the addition of 0.5% (*w*/*v*) bromocresol purple solution (provided by Beijing Aobo Star Biotechnology Co., Ltd.) at the ratio of 1.5% (*v*/*v*). Culture media were all sterilized at 115 °C for 20 min in an autoclave. The 1% (*v*/*v*) activated bacterial cultures were added into the prepared medium containing different carbohydrates respectively and incubated anaerobically at 30 °C for 24 to 48 h. The control medium (without bacterial inoculum) was prepared using the same method. Color changes were observed at 48 h to evaluate the carbohydrate utilization. The growth characteristics strain *F. sanfranciscensis* Fs_1010 with high EPS yield (202.341 mg/L) using different carbohydrate as carbon source were further studied. For this purpose, the autoclaved mMRS medium was prepared without mono and disaccharides, supplemented with sterile filtered 2 g/100 mL of either glucose, fructose, maltose, and glucose + fructose, respectively. After inoculation at 1% (*v*/*v*), the strain was incubated at 30 °C for 48 h while growth was followed by measuring the absorbance values at 600 nm (A_600_ nm). Concomitantly, to determine the rate of acid production, pH and TTA were measured according to Zhang et al. (2020) [21].

### 2.5. Genotypic Analysis of EPS-Producing Strains

EPS-producing ability of the 14 *F. sanfranciscensis* strains was assayed via the phenol–sulfuric acid method [22]. Functional annotation information and blast alignment were used to predict and analyze the genes contributed to EPS synthesis in 14 *F. sanfranciscensis* strains (e-value was <1 × 10^−5^). The Origin Pro 2024 software was used to create a heat map.

### 2.6. Genotypic and Phenotypic Analysis of Antibiotic Resistance

The antibiotic resistance genes in 14 *F. sanfranciscensis* strains were analyzed using the CARD database (Comprehensive Antibiotic Resistance Database, http://arpcard.Mcmaster.ca, accessed on 25 December 2023). If the sequence matching degree of the resistance gene reaches 20% (e-value < 1 × 10^−5^), the antibiotic resistance gene is considered to exist. The Origin 2023 software was used to build a thermal map of the predicted data.

Based on the American Association for Clinical and Laboratory Standards Institute (CLSI) guidelines [34], the antibiotic resistance of the 14 *F. sanfranciscensis* strains was tested using the disk diffusion method. The antibiotics per disk were as follows: gentamicin, kanamycin, streptomycin, erythromycin, clindamycin, benzathine, ampicillin, tetracycline, chloramphenicol, and mitomycin-sulfamethoxazole, purchased from Liofilchem. The strains that were classified as susceptible (S, zone diameter > 20), intermediate (IR, 15 < zone diameter < 19), or resistant (R, zone diameter ≤ 14) hinged on the diameter of the zone of inhibition around the disk [30].

### 2.7. CRISPR Identification and Bacteriocin-Derived Genes Analysis

CRISPRCas Finder [35] (https://crisprcas.i2bc.paris-saclay.fr/CrisprCasFinder/Index, accessed on 1 January 2024) was used to predict the genome of CRISPR/Cas system, including the sequence of DRs (repeat sequence), Spacers (spacer region), and Cas proteins via WebLogo (http://weblogo.berkeley.edu/) for visual conservative repetitive sequencing directly. DRs RNA secondary structure via RNAfold (http://rna.tbi.univie.ac.at/cgi-bin/RNAWebSuite/, accessed on 5 January 2024) was used to forecast the default parameters. Using online tools BAGEL4 (http://bagel5.molgenrug.nl/, accessed on 6 January 2024), the bacterium strain gene prediction was compared to find assume operon gene cluster information.

## 3. Results

### 3.1. Genome Characteristics of F. sanfranciscensis

Genomic information of all the above-mentioned strains is listed in Table 1. In this study, the genome size of 45 *F. sanfranciscensis* strains ranged from 1.26 Mbp to 1.37 Mbp, with an average of 1.31 Mbp. Similarly, the average value of GC content was 34.25%, ranging from 33.07% to 35.20%. The genome was predicted to have an average number of 1274 protein-coding sequences (CDS) that ranged from 1184 to 1386.

### 3.2. Pan-Genome and Core Genes of F. sanfranciscensis

The basic genomic features of *F. sanfranciscensis*, pan-genome and core genome analyses were further investigated. The genomic dynamic characteristics between the number of core genes and pan-genes and the sequenced strains are depicted (Figure 1A). When the number of *F. sanfranciscensis* strains increased, the quantity of pan-genomes expanded further, whereas the core-genome remained at a steady level. After adding the 45th genome, the pan-genome appeared stable with 4494 genes, whereas there were only 159 genes in the core genome. However, the exponent γ value was less than zero, meaning the pan-genome of *F. sanfranciscensis* was closed. A Venn diagram (Figure 1B) examines the specific core genes and accessory genes of the 45 different *F. sanfranciscensis* strains. The 45 *F. sanfranciscensis* strains used in this research had an average total number of homologous gene clusters of 1307 with 12.40% core genes with 0 to 51 unique genes per strain. In 14 *F. sanfranciscensis* strains, the size of core genes, or 0.70 Mbp of 1.70 Mbp, accounted for 41.18% of the pan-genome size. The functions of the core genes of 45 *F. sanfranciscensis* are mainly related to nucleotide metabolism, energy production and conversion, lipid transport and metabolism, amino acid metabolism, and carbohydrate metabolism (Figure 1C). Among them, 49.41% (3514/7112) functional genes were related to translation, ribosomal structure and biogenesis, 2.53% (180/7112) were related to amino acid transport and metabolism, and 1.90% (135/7112) were related to carbohydrate transport and metabolism. Furthermore, 4.43% (315/7112) unknown function proteins were discovered in 45 *F. sanfranciscensis* strains.

### 3.3. ANI and Phylogenetic Analysis of F. sanfranciscensis

Strains should be considered as the same species when the ANI value exceeds 95% [30,44]. To further explore the relatedness of *F. sanfranciscensis* strains, the ANI values of 45 *F. sanfranciscensis* strains were analyzed. Figure 2A shows that the calculation of the ANI values of all 45 strains resulted in 99.75% to 100% similarity. The results showed that all the 45 strains were *F. sanfranciscensis* and there were no subspecies. Phylogenetic analysis divided the 45 strains into three major genetic clusters based on homologous genes (Figure 2B). In contrast to the other 43 *F. sanfranciscensis* strains, the strain of *F. sanfranciscensis* Gs2 and Gs9 was distributed into two different branches, respectively.

### 3.4. Genotypic and Phenotypic Analysis of Carbohydrate Metabolism

According to gene functional characterization, numerous genes were involved in carbohydrate transport and metabolism in all strains. Figure 3A,B show that 23 genes encoding carbohydrate-active enzymes were present in the 14 *F. sanfranciscensis* strains, respectively. The CAZyme-coding genes mainly belonged to glycosyltransferase families (GTs), glycoside hydrolases families (GHs), and carbohydrate esterases (CEs) families, which are responsible for the formation of glycosidic bonds, hydrolysis of rearranged glycosidic bonds, and hydrolysis of carbohydrate esters, respectively. These genes reflect the potential of LAB to synthesize and hydrolyze different carbohydrates. Among them, GT4 (sucrose synthase; EC 2.4.1.13), GH25 (lysozyme; EC 3.2.1.17), GH65 (α, α-alglucan; EC 3.2.1.28), GH109 (α-N-acetylgalactosaminidase; EC 3.2.1.49), and CE10 (carboxyl esterase; EC 3.1.1.3) were more abundant in all the 14 *F. sanfranciscensis* strains.

By analyzing the 14 *F. sanfranciscensis* strains’ abilities to utilize carbohydrates, we aimed to determine whether the genotypic classification corresponded to phenotypic characteristics. The results showed that all of these strains were able to utilize D-maltose by expressing genes belonging to family GH65 (maltose phosphorylase; EC 2.4.1.8), but did not utilize soluble starch. In addition, there were differences in the ability of *F. sanfranciscensis* strains to utilize D-glucose (85.71%), D-trehalose (78.57%), and D-fructose (42.86%). It is worth mentioning that all 14 *F. sanfranciscensis* strains were unable to utilize common carbon sources in sourdough, such as D-sucrose, although they had the genes of *malZ* (α-glucosidase; EC:3.2.1.20) which hydrolyzes sucrose to release D-fructose and D-glucose as an endocellular enzyme. Although GH70 (alternate sucrase; EC 2.4.1.140) were present in *F. sanfranciscensis* (Fs_1003, Fs_1004, Fs_1005, Fs_1008, Fs_10010) indicating their potential to metabolize sucrose efficiently, they failed to metabolize sucrose due to the lack of corresponding sucrase genes (such as sucrose PTS permease, EC 2.7.1.211; sucrase–isomaltase, EC 3.2.1.48 3.2.1.10; etc.). In addition, none of the 14 *F. sanfranciscensis* strains could utilize these carbohydrates, including D-lactose, D-galactose, L-arabinose, and mannitol.

The above results showed that all 14 *F. sanfranciscensis* strains could metabolize maltose. However, there was strain variability in the utilization of D-glucose, D-fructose, and D-trehalose (Figure 3C). Notably, the D-glucose utilization capacity of *F. sanfranciscensis* Fs_1010 were different between the carbohydrate utilization phenotype test (which did not utilize glucose) (Figure 3C) and the growth curve test (which utilized glucose but grew more slowly than other carbon sources) (Figure 3D). The growth density of Fs_1010 during 4 to 12 h of fermentation was the frontrunner when grown in a medium containing maltose (as a single carbon source) than in a medium containing only three other carbon sources. After 12 h of fermentation, the growth density of strain Fs_1010 was relatively higher on fructose as a single carbon source compared to its growth on glucose or a mixture of glucose and fructose. These results were consistent with the growth features of *F. sanfranciscensis*, preferring fructose-rich substrates.

Acidification is important for improving the characteristics of baked products [45]. The ability of strain Fs_1010 to produce acid during fermentation with different carbon sources was further assessed, as shown in Figure 3E. The results of pH and TTA showed that the acid levels produced by strain Fs_1010 varied with carbon sources. When maltose and fructose were used as the only carbon sources for the fermentation solution for 48 h, the final pH of the solution reached 4.0, but the pH of the medium containing glucose decreased only to 4.6.

### 3.5. Genotypic Analysis of EPS

The EPS made by *F. sanfranciscensis* are recognized to improve the texture, rheology, and shelf-life of bread [3]. In this research, the biosynthesis genes and production of EPS were analyzed. Genes related to EPS biosynthesis were annotated using the NCBI database (https://www.ncbi.nlm.nih.gov/, accessed on 5 March 2024). The genes of *epsA*, *epsB* and *epsD* have been identified, which are responsible for encoding LytR-transcription regulatory factor, tyrosine protein kinase, and tyrosine protein phosphatase respectively (Table 2). These *eps* genes play an important role in gene expression, signaling pathway regulation and biological activity functions during EPS synthesis. In addition, *wzy* and *wzx* genes were identified. The *wzy* gene encodes a polysaccharide polymerase, which catalyzes the polymerization of sugar residues and extends and forms EPS. The *wzx* gene promotes the assembly and extracellular transport of EPS. The genomic analysis also revealed many genes encoding glycosyltransferase (GTF) for biosynthesis of EPS repeat units. Other genes related to EPS synthesis in 14 *F. sanfranciscensis* strains are displayed in Figure 4B. The results revealed that the distribution of the gene of glycosyltransferase was significantly different, whereas the composition of chain-length determining protein and transcriptional regulator was conservative.

Then, we also analyzed the carbohydrate-active enzymes genes related to EPS production across the 14 *F. sanfranciscensis* strains. The result showed that the production of EPS and the distribution of carbohydrate enzyme genes were different (Figure 4A, Appendix A). The genes of GH68 (levansucrase; EC 2.4.1.10) and GH53 (endo-β-1,4-galactanase; EC 3.2.1.89) family were only present in *F. sanfranciscensis* strains Fs_1007 and Fs_1009, which might suggest that Fs_1009 and Fs 1007 strains were more adept than other strains in sucrose utilization. The high EPS production, 191.9 mg/L, by strain Fs_1009 could be related to its ability to produce levansucrase (Figure 3B), thanks to the gene that encodes inulosucrase (EC:2.4.1.9), which can metabolize sucrose to inulin. The strain Fs_1010, reaching up to 202.3 mg/L of EPS contains glycoside hydrolase family 70 (GH70) genes (dextransucrase; EC 2.4.1.5) that regulate glucan production. The genes from GT5 (α-1,3-glucan synthase; EC 2.4.1.183) only exist in Fs_1003 out of all the 14 *F. sanfranciscensis* strains.

### 3.6. Phenotypic and Genotypic Analysis of Antibiotic Resistance

Figure 5A shows that 14 *F. sanfranciscensis* strains had antibiotic resistance genes ranging from 65 to 71. Strain Fs_1002 appeared to contain the fewest antibiotic resistance genes, while Fs_1013 turned out to have the most. The putative antibiotic resistance genes were annotated to 51 antibiotic resistance genes in 14 *F. sanfranciscensis* strains. All the 14 *F. sanfranciscensis* strains in our trial contained the types of antibiotic resistance genes of peptide, such as *bcrA*; macrolide, such as *macB*; lincosamide, such as *lmrD*; rifamycin, such as *efrA*; tetracycline, such as *tetT*; and glycopeptide, such as *vanHD* and *vanHO* (Figure 5B,C). *F. sanfranciscensis* Fs_1001, Fs_1002, and Fs_1013 strains had the unique distribution of resistance genes, each containing one unique gene, *tva* (*A*), *Erm* (49), and *tet* (44), respectively, while Fs_1002 was the only one of the 14 strains without *ErmQ* resistance genes. The *dfrC* and *vatB* genes were only present in Fs_1007 and Fs_1009, and *tetA* (46) was only present in Fs_1008 and FS_1010. However, compared with other *F. sanfranciscensis* strains, Fs_1001, Fs_1002, and Fs_1013 did not contain the three resistance genes *dfrC*, *vatB*, and *tetA* (46). The antibiotic resistance of the 14 *F. sanfranciscensis* strains was further determined using the disk diffusion method. As shown in Table 3, the 14 *F. sanfranciscensis* strains were sensitive to erythromycin (E), clindamycin (CD), ampicillin (AMP), and chloramphenicol (C). However, it showed resistance towards aminoglycosides gentamicin (CN), kanamycin (K), streptomycin (S), and to Trimethoprim-sulfamethoxazole (SXT).

In addition, 14 *F. sanfranciscensis* strains showed different levels of resistance towards penicillin; oxacilin OXO (low susceptibility) and ampicillin AMP (high susceptibility). Even though tetracycline resistance genes *tetA* (58), *tetT*, and *tetB* (58) were present in all strains, still these three species showed different levels of resistance towards tetracycline TE. Through the CRAD database analysis, the susceptibility of these strains to various antibiotics exhibited substantial variations; the following scenario will explain. Few strains exhibited resistance and had genes corresponding to antibiotic resistance genes that showed resistance (for instance, all of these strains contained *Staphylococcus aureus LmrSdu* resistance gene that showed resistance to aminoglycosides antibiotics, CK, K, and S). Few strains had the antibiotic resistance phenotype but no associated resistance genes, for instance, all the 14 *F. sanfranciscensis* strains were resistant to SXT, but no sulfonamide resistance gene could be found. There is another possibility that strains with resistance genes may not exhibit the resistance phenotype. For instance, 14 *F. sanfranciscensis* strains contained *catB11* gene that conferred chloramphenicol resistance but these strains were not resistant to chloramphenicol, suggesting that the gene *catB11* was probably not the main gene for chloramphenicol resistance. Moreover, all the 14 *F. sanfranciscensis* strains contained gene *macB* that encodes an ABC transporter protein, but the phenotypic test results of all strains did not show resistance to erythromycin (E).

### 3.7. CRISPR Identification of F. sanfranciscensis

The CRISPR-Cas systems of genome of 45 *F. sanfranciscensis* strains were analyzed. All 45 strains contained CRISPR sequences, CRISPR sequences without Cas proteins were ignored due to its inability to silence foreign DNA. Therefore, of the 45 *F. sanfranciscensis* strains, 42 strains contained complete CRISPR-Cas systems (Appendix A). The most common Cas gene cluster was Type ⅡA (92.86% of strains), which mainly contained cas9, cas1, cas2, and csn2 genes (Figure 6A). The results showed that Type ⅡA was prevalent in *F. sanfranciscensis*. In addition, Type IE was also found in 6 *F. sanfranciscensis* strains (Appendix A). Remarkably, *F. sanfranciscensis* TMW 1.640, Gs2 and Fs_10001 contained both Type ⅡA and Type IE CRISPR-Cas systems (Figure 6A,B). In addition, the RNA secondary structure and the minimum free energy (MFE) of DR sequences were predicted using the RNAfold tool. The results revealed that the secondary structure could be divided into three types of typical structures (Figure 6). The first type was a circular structure (0.00 kcal/mol), which may not have the active ability to silence foreign protein expression; this needs further confirmation. The second structure consisted of a circular structure at the head and tail and a stem-like structure in the middle (−2.70 kcal/mol). Compared with the second structure type, the middle stem structure of the third structure also had the characteristics of conserved RNA secondary structure, i.e., containing one or two small rings and G:U base pairs, corresponding to the minimum free energy of −10.10 kcal/mol and −5.70 kcal/mol, respectively. Such structures are important for CRISPR function.

Furthermore, the genomes of 45 *F. sanfranciscensis* strains were analyzed to determine the ability of the CRISPR-Cas system to protect against foreign invasion. Among the 45 genomes, 45 phage regions were found, whereas 10 strains had no phages annotated. There were two complete phage regions, one suspicious region, and 42 incomplete regions (Appendix A). Only two strains (TMW 1.1597 and TMW 1.907) had complete phage regions, meaning that only 4% of the strains had functional phages. Nonetheless, these results still require further research to better understand the role of the CRISPR-Cas system in *F. sanfranciscensis*.

### 3.8. Bacteriocin Operons Analysis in F. sanfranciscensis

Using the BAGEL4 tool, the bacteriocin operons of 45 *F. sanfranciscensis* strains were predicted. There were seven bacteriocin operons, five of which have no transporter and one core peptide that has not yet been described (Figure 7). Therefore, these five bacteriocin operons were considered incomplete.

The pediocin PA-1 operon was found in the genomic plasmid sequences of *F. sanfranciscensis* Fs_1001, Fs_1004, and Fs_1005. The monopeptidycin pediocin PA-1 is one of class IIA bacteriocins showing strong activity against Listeria monocytogenes. In addition to pediocin PA-1, Lacticin_3147_A1 was also found in *F. sanfranciscensis* Ah4 strain. These findings suggest that Ah4 strains have antimicrobial activity potentiality owing to the bacteriocin operons. However, because some operons were incomplete or core peptides have not yet been described, further research is needed to understand their detailed properties and functions.

## 4. Discussion

*Fructilactobacillus sanfranciscensis* is a key bacterial species of the sourdough microbiota which plays an essential role in improving the aroma, texture, and nutritional characteristics of traditional Type I sourdough. Several strains of *F. sanfranciscensis* with different phenotypic and genotypic characteristics have been reported from various sourdough samples from diverse geographical origins [11]. Moreover, the beneficial effects of *F. sanfranciscensis* on sourdough is a strain-dependent characteristic [3]. Many phenotypic characteristics of *F. sanfranciscensis* strains are of industrial interest, acid production capacity, antibacterial capacity, metabolites, etc. Primarily its carbohydrate metabolism has attracted much attention. It is imperative to discriminate strains of *F. sanfranciscensis* using comparative genomics to formulate effective starter cultures for sourdough-based bread. *F. sanfranciscensis* strains with a relatively higher ability to ferment maltose are competitive against other strains, showing inverse behavior. *F. sanfranciscensis* TMW 1.392 was the most common strain due to its ability to utilize different carbohydrates and exploit electron acceptors like oxygen and fructose compared with other *F. sanfranciscensis* strains in the research of Rogalski et al. (2021) [17].

Whole genome sequences of only a few strains of *F. sanfranciscensis* are available, limiting the possibilities of genome-based comparative studies. Complete genome sequences are important for the precise description of core genomes, for the identification of strain-specific genes, and for the effective selection of important functional traits among a large number of strains. Therefore, this study analyzed the genomes of 14 *F. sanfranciscensis* strains isolated from sourdough samples and 31 genomes acquired from the NCBI database. Genotypic and phenotypic characteristics in terms of carbohydrate metabolism and antibiotic resistance were compared and combined with comparative genomics to analyze similarities and differences among the strains.

The small genome size of *F. sanfranciscensis* indicates that it participates in a limited number of chemical reactions, which can be linked to its successful adaptation towards the sourdough niche. According to streamlining theory in microbial ecology, bacteria with small genomes effectively use nutrients in populations when the adequate population size is large and nutrients limit growth [46]. Microbial niches where adaptation success is determined by resource competition, the phrase of Leonardo da Vinci fits in: ‘simplicity is the ultimate sophistication.’ *F. sanfranciscensis* has the smallest genome among all lactobacilli, and it has the highest density of ribosomal RNA operons per Mbp, when compared with all known genomes of free-living bacteria in nature [3,11,42]. The aforementioned features are associated with the rapid growth characteristics of the organism [42]. The findings of genomic functional annotation analysis revealed that the large multifunctional genes of *F. sanfranciscensis* were primarily involved in nucleotide metabolism, energy production and conversion, amino acid metabolism, and carbohydrate metabolism. However, the functional landscape of an entire set of genes has not yet been unveiled because 4.43% of gene functions still need to be determined.

*Fructilactobacillus sanfranciscensis* strains are known to have variability in genes related to carbohydrate metabolism [11]. This study revealed that the strains had a significant variability in phenotypic characteristics related to carbohydrate metabolism, rather than in genes related to carbohydrate utilization. Based on the carbohydrate utilization phenotypes of 14 *F. sanfranciscensis* strains, they can be divided into two categories (Figure 3C). There are two groups strains, one can use both maltose and glucose, while another group can only use maltose. The results of this study agree with the result of Rogalski et al. (2020a) [16] and Foschino et al. (2001) [25]. Moreover, 14 *F. sanfranciscensis* strains were divided into two major clusters based on the family genes number of GT4 and GT2_Glycos_transf_2 (Figure 3B). In this study, 14 *F. sanfranciscensis* strains were found to utilize maltose, and contained genes of maltose phosphorylase *map* and phosphoglucomutase *pgm*. In this study, all 14 *F. sanfranciscensis* strains contained genes for phosphoketolase pathway from maltose, one of the key carbohydrates in the bread, to produce energy and glucose. And one of the main reasons that *F. sanfranciscensis* can thrive in sourdough is its ability to utilize maltose more efficiently. This characteristic is also linked to the ability of *F. sanfranciscensis* to establish a symbiotic relationship with *S. cerevisiae*, resulting in better adaptation to the sourdough environment [11,17]. There have also been reports of strain-specific interactions between *F. sanfranciscensis* and the yeasts [17]. *S. cerevisiae* metabolizes sugars during fermentation to produce significant levels of CO_2_, which improves the bread volume and crumb texture. Moreover, yeast creates a favorable environment for the growth of anaerobic LAB, such as *F. sanfranciscensis*. At the same time, *F. sanfranciscensis* can hydrolyze maltose and proteins through its metabolism to provide glucose and amino acids to baker’s yeast for its enhanced growth [47].

The result may be due to the fact that *F. sanfranciscensis* takes longer to adapt to growing on glucose, as the genetic setting should allow for all strains that exploit glucose [11,48], such as the genes of *glk*. In addition, fructokinase exists in all 14 *F. sanfranciscensis* strains, but there are still eight strains cannot utilize fructose (Figure 3C). The reason for this incapability is the gene of fructokinase in these eight strains which may be not functional. In the literature, it is described that fructose is favored by *F. sanfranciscensis* for use as an external electron acceptor to increase energy and speed up growth [11,49]. The variations in the sourdough fermentation behavior can be largely attributed to strain-species differences in carbohydrate metabolism of *F. sanfranciscensis*.

The EPS of LAB have high industrial value, can significantly improve the texture and rheological properties of fermented food, including antioxidant, antibacterial, immune regulation, and other biological activities [50]. Via the analysis of carbohydrate enzyme genes and genes related to EPS synthesis, this article provides valuable insights for the analysis of EPS biosynthesis of *F. sanfranciscensis*. We identified multiple EPS biosynthesis genes and carbohydrate enzyme genes, including flippase, polymerase, and GTs in 14 *F. sanfranciscensis* strains. The identification of these genes contributes to further understanding of the molecular mechanisms and regulatory networks involved in the EPS production of *F. sanfranciscensis*, which has far-reaching implications for optimizing EPS production, modifying EPS structure, and exploring the broad potential of EPS applications.

Globally, food safety is becoming an increasingly important topic while exploring the LAB’s probiotic or food fermentation potential [51]. Therefore, the presence of antibiotic resistance genes and their dissemination potential should be thoroughly evaluated before declaring a bacterial strain safe. Many studies have determined antibiotic resistance of bacteria from fermented foods and probiotic formulations [52,53,54]. The antibiotic susceptibility assessment in this study revealed that all the 14 *F. sanfranciscensis* strains included multidrug antibiotic resistance patterns. A total of 14 *F. sanfranciscensis* strains were sensitive to erythromycin (E) and clindamycin (CD), but resistant to aminoglycosides (streptomycin (S), kanamycin (K), and gentamicin (CN), as also presented in the description of current research [34,55]. Membrane impermeability is considered the primary mechanism of aminoglycoside resistance in LAB due to the lack of a cytochrome-related electron transport system that promotes drug uptake [34]. All 14 *F. sanfranciscensis* strains contained at least one aminoglycoside resistance gene (such as *Staphylococcus aureus LmrS*, *baeS*, *baeR*, *aadA23*, *kdpE*), and aminoglycoside antibiotic genes mainly through the efflux of antibiotics to antibiotic resistance. In addition, all 14 *F. sanfranciscensis* strains contained the *macB* gene. However, they did not produce resistance to macrolide antibiotics (erythromycin), revealing that *macB* was not a crucial gene for erythromycin resistance [30]. It was speculated that this resistance gene could be silent in these strains. The *macB* gene did not have high sequence similarity with the CRAD database (<44.9%). Antibiotic resistance in LAB may be impacted by the expression of antibiotic resistance genes. The 14 strains of *F. sanfranciscensis* showed various degrees of resistance to the tetracycline analog tetracycline TE. Despite corresponding resistance genes, no risk of horizontal transfer was found in mobile elements such as prophages. Therefore, we assumed that these strain did not pose a safety risk concerning the function or dissemination of gene *macB*. So, all *F. sanfranciscensis* strains used in this research were safe regarding antibiotic resistance and can be further applied to make sourdough-based products.

The CRISPR-Cas (clustered regularly interspaced short palindromic repeats, CRISPR-associated proteins) system offers acquired resistance against invasive elements such as phages and plasmids in most bacteria [56,57]. CRISPR consists of a number of short, conserved repeat regions and spacers. Cas, found near the CRISPR site, is a double-stranded DNA nuclease that cuts the target site. Cas functions as a cluster of three to more than 10 genes, which can be divided into six types (I to VI) and more than 30 subtypes [58]. There is species-specific in the distribution of Cas gene clusters. For instance, Type ⅡA mainly exists in the CRISPR system of *Pediococcus pentosus*, *Latilactobacills curvatus*, and *Lactobacillus reuteri*. The TypeIE Cas gene cluster mainly exists in *Lactobacillus acidophilus* and *Lactobacillus ferment*. The genome sequence of *Lactobacillus* generally contains Type II CRISPR-CAS system [59], which is naturally active and effectively targets invasive and genomic DNA. There were 31 strains of *F. sanfranciscensis* with Type IIA CRISPR-Cas system containing the cas gene (Appendix A); Cas9 has the function of targeting and cutting the viral genome and can mediate genome editing. Therefore, *F. sanfranciscensis* can be used as a candidate for gene editing and provides the basis for cracking lysed phages in the food industry.

Bacteriocins in LAB have great biosafety and wide industrial application [60]. For example, bacteriocins can be used to develop new types of food biological preservatives due to its natural antibacterial activity [61]. In addition, bacteriocins have the potential to tailor host flora and control foodborne pathogens as the relatively narrow and specific killing spectrum [62]. Bacteriocins can also be used as an alternative to antibiotics in animal feed additives [63]. In this research, *F. sanfranciscensis* Ah4 could be considered the potential candidate as a source of bacteriocins. However, due to the regulatory complexity of bacteriocins in the production process, and that their biosynthesis and transport mechanisms are not fully understood, further research should provide enough experimental evidence to clarify the mechanism and promote upscale industrial application.

## 5. Conclusions

A comparative genomic analysis of 45 strains of *F. sanfranciscensis* revealed that the pan-genome of this species was closed. Moreover, the strain-specific differences were primarily reflected in carbohydrate utilization, EPS biosynthesis, antibiotic resistance, and immune/competition related factors (CRISPR-Cas and bacteriocins operon). Additionally, the CRISPR/Cas system was predicted to be of Type IIA and Type IE. There were two bacteriocins operon identified, including pediocin PA-1 and Lacticin_3147_A1. As a result of genotypic and phenotypic characterization, the biotechnological potential of *F. sanfranciscensis* strains can be further explored and the derived knowledge can be used to develop standardized sourdough starter culture.

## Figures and Tables

**Figure 1 microorganisms-12-00845-f001:**
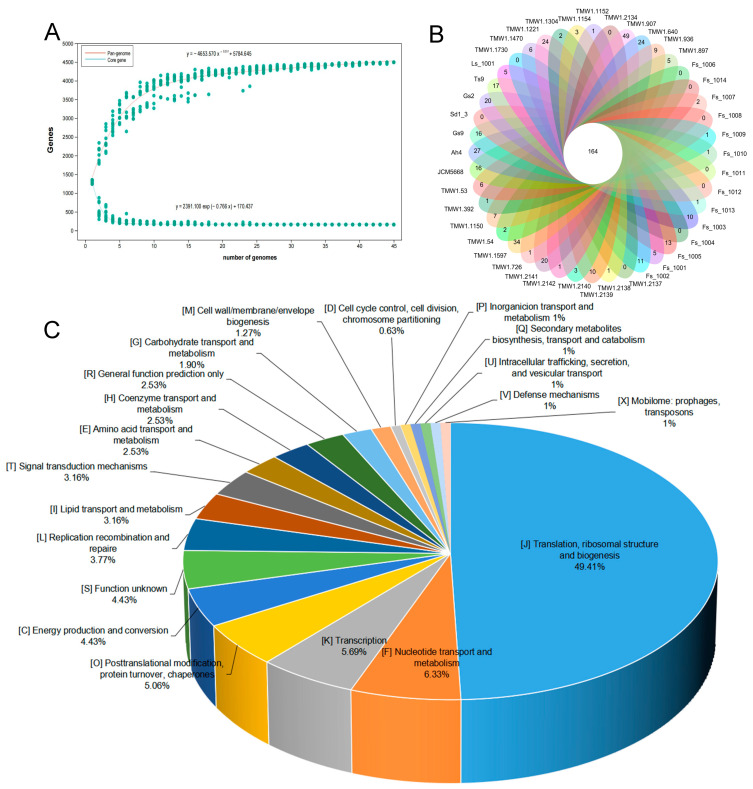
Pangenome and core genes of *Fructilactobacillus sanfranciscensis*. (**A**) Pangenome and core genes of 45 *F. sanfranciscensis* strains. (**B**) Venn diagram showing the unique and core genes. (**C**) The function annotation of core genomes based on COG database.

**Figure 2 microorganisms-12-00845-f002:**
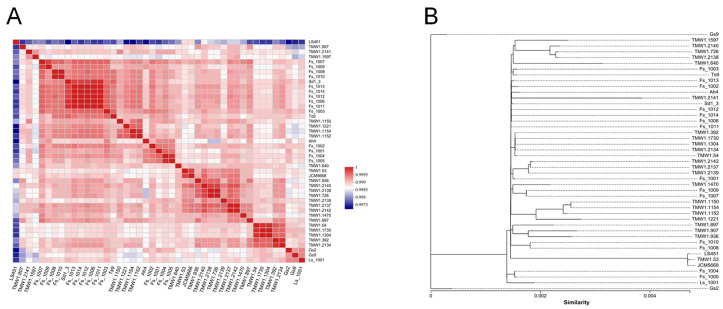
ANI and phylogenetic analysis of *Fructilactobacillus sanfranciscensis*. (**A**) Heat map of 45 strains of *F. sanfranciscensis* via ANI-TETRA. (**B**) Phylogenetic tree of 45 *F. sanfranciscensis* genomes based on homologous genes.

**Figure 3 microorganisms-12-00845-f003:**
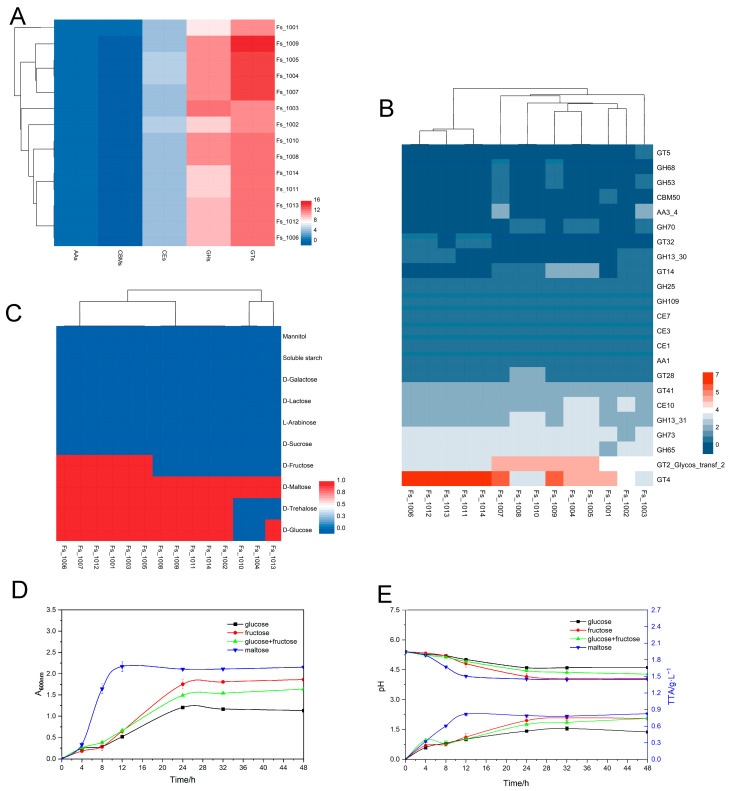
Comparative carbohydrate utilization genotype-phenotype analysis of 14 strains of *Fructilactobacillus sanfranciscensis*. (**A**): Heat map of gene number distribution of five carbohydrate-active enzymes in different strains. (**B**): Heat map of genes associated with carbohydrate-active enzymes in strains. (**C**): Carbohydrate utilization capacity of different strains. (**D**): Growth curve of *F. sanfranciscensis* Fs_1010 on different carbon sources. (**E**): Acid production performance of *F. sanfranciscensis* Fs_1010 on different carbon sources, The declining trend line graph represents the changes in pH at different fermentation times. The upward trend line graph represents the changes in total acidity (TTA) at different fermentation times.

**Figure 4 microorganisms-12-00845-f004:**
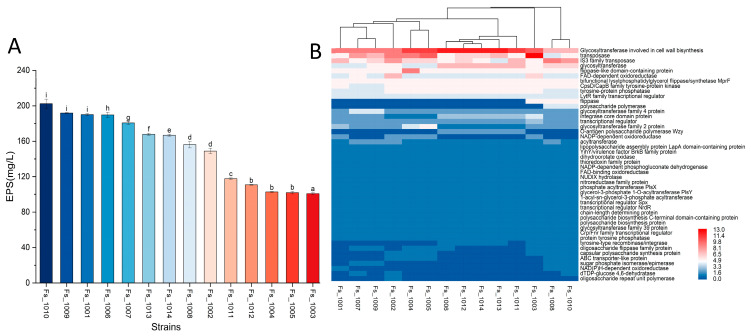
The exopolysaccharides (EPS) production and genes associated with regulating EPS synthesis of 14 *Fructilactobacillus sanfranciscensis* strains. (**A**) The EPS production of 14 *F*. *sanfranciscensis* strains. Different lowercase letters (a–i) indicated a significant difference (*p* < 0.05). (**B**) Heat map of genes associated with synthetic EPS genes in 14 *F*. *sanfranciscensis* strains.

**Figure 5 microorganisms-12-00845-f005:**
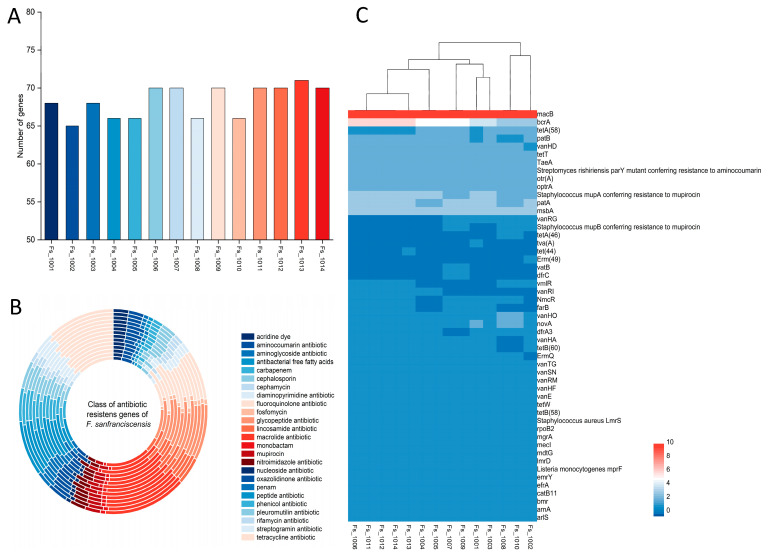
Comparative genotypic and phenotypic analysis of antibiotic resistance in 14 strains of *Fructilactobacillus sanfranciscensis*. (**A**) Number of resistance genes present in different strains. (**B**) Classification of antibiotic resistance genes of 14 strains of *F. sanfranciscensis* strains, from inside to out, in order of strain number from smallest to largest. (**C**): Heat map analysis of antibiotic resistance genes of different strains.

**Figure 6 microorganisms-12-00845-f006:**
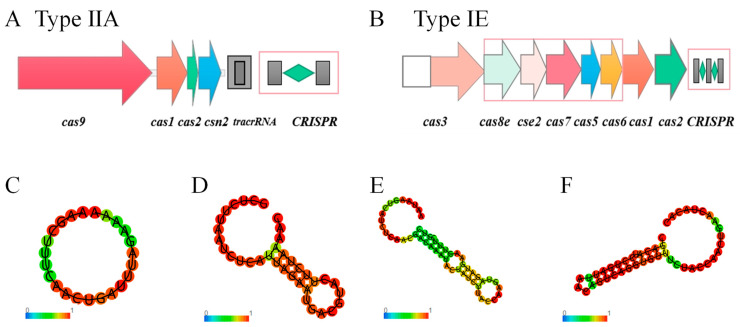
Prediction of Cas protein and RNA secondary structure of CRISPR DR in *Fructilactobacillus sanfranciscensis*. (**A**,**B**) are the type of the CRISPR/Cas locus of *F. sanfranciscensis*, respectly. The prediction of RNA secondary structure of CRISPR DR in *F. sanfranciscensis* (**C**–**F**).

**Figure 7 microorganisms-12-00845-f007:**
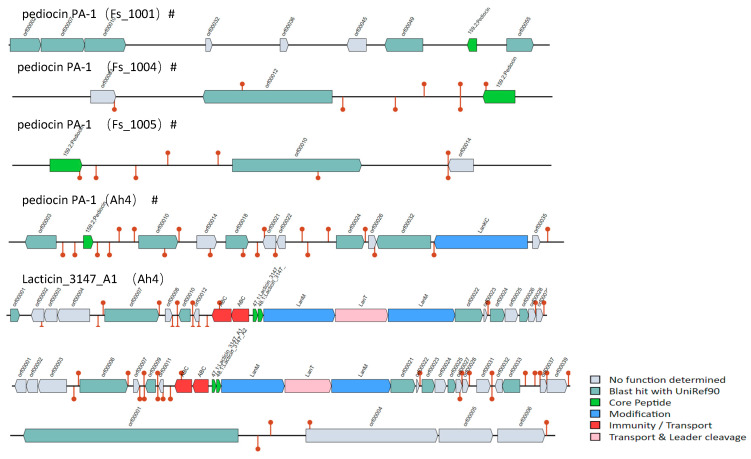
The potential bacteriocin operons identified in *Fructilactobacillus sanfranciscensis*. “#” indicate lack of core peptide and transporter, respectively.

**Table 1 microorganisms-12-00845-t001:** Information of 45 strains of *Fructilactobacillus sanfranciscensis*.

Organism	Strain	Accession No.	Isolation Source	Genome Size (bp)	GC Content (%)	CDS Coding (Total)	References
*Fructilactobacillus sanfranciscensis*	DSM20541; TMW 1.53	MIYJ00000000	Sourdough, USA; San Francisco sourdough	1,295,539	34.76	1221	[36]
*Fructilactobacillus sanfranciscensis*	TMW 1.54	NZ_MIYE01000000	Rye sourdough, Germany	1,308,584	34.62	1225	[37]
*Fructilactobacillus sanfranciscensis*	TMW 1.392	NZ_MIYH01000000	Sourdough, Freising Germany	1,262,093	34.49	1185	[38]
*Fructilactobacillus sanfranciscensis*	TMW 1.640	SCEZ00000000	Wheat sourdough, Switzerland	1,297,108	34.85	1243	[39]
*Fructilactobacillus sanfranciscensis*	TMW 1.726	NZ_MIYD01000000	Sourdough, Italy	1,253,149	34.67	1184	[40]
*Fructilactobacillus sanfranciscensis*	TMW 1.897	SCEP00000000	Sourdough, Greece; Athens	1,249,687	34.60	1209	[41]
*Fructilactobacillus sanfranciscensis*	TMW 1.907	SCEY00000000	Sourdough, Greece; Athens	1,281,500	34.69	1257	[16]
*Fructilactobacillus sanfranciscensis*	TMW 1.936	SCEX00000000	Sourdough, Greece; Athens	1,242,512	34.74	1190	[16]
*Fructilactobacillus sanfranciscensis*	TMW 1.1150	NZ_MIYG01000000	Sourdough, Germany	1,281,791	34.69	1195	[16]
*Fructilactobacillus sanfranciscensis*	TMW 1.1152	SCEV00000000	Sourdough, USA	1,253,076	34.58	1186	[16]
*Fructilactobacillus sanfranciscensis*	TMW 1.1154	SCEU00000000	Sourdough, USA	1,242,516	34.52	1191	[16]
*Fructilactobacillus sanfranciscensis*	TMW 1.1221	SCET00000000	Sourdough, France	1,256,830	34.48	1203	[16]
*Fructilactobacillus sanfranciscensis*	TMW 1.1304	SCES00000000	Rye sourdough, Germany	1,309,936	34.59	1280	[42]
*Fructilactobacillus sanfranciscensis*	TMW 1.1470	SCER00000000	Sourdough, Russia	1,264,120	34.55	1211	[16]
*Fructilactobacillus sanfranciscensis*	TMW 1.1597	NZ_MIYF01000000	Rye sourdough, Germany	1,318,230	34.81	1232	[16]
*Fructilactobacillus sanfranciscensis*	TMW 1.1730	SCEQ00000000	Sourdough, Germany	1,307,874	34.60	1279	[16]
*Fructilactobacillus sanfranciscensis*	TMW 1.2137	NZ_MIXX01000000	Sourdough, Italy	1,272,753	34.73	1210	[43]
*Fructilactobacillus sanfranciscensis*	TMW 1.2138	NZ_MIXY01000000	Sourdough, Italy	1,252,720	34.68	1190	[43]
*Fructilactobacillus sanfranciscensis*	TMW 1.2139	NZ_MIXZ01000000	Sourdough, Italy	1,329,228	34.77	1256	[43]
*Fructilactobacillus sanfranciscensis*	TMW 1.2140	NZ_MIYA01000000	Sourdough, Italy	1,293,144	34.71	1201	[43]
*Fructilactobacillus sanfranciscensis*	TMW 1.2141	NZ_MIYB01000000	Sourdough, Italy	1,309,594	34.71	1241	[43]
*Fructilactobacillus sanfranciscensis*	TMW 1.2142	NZ_MIYC01000000	Sourdough, Italy	1,278,409	34.71	1212	[43]
*Fructilactobacillus sanfranciscensis*	TMW 1.2134	SCEW00000000.1	Rye sourdough, Germany	1,254,138	34.46	1230	[16]
*Fructilactobacillus sanfranciscensis*	LS451	NZ_CP045563.1	San Francisco sourdough; South Korea	1,310,991	35.15	1318	(Korea University; 3 November 2019)
*Fructilactobacillus sanfranciscensis*	JCM 5668	NZ_QRFO00000000	San Francisco sourdough	1,267,448	34.82	1318	(ChunLab; 2 August 2018)
*Fructilactobacillus sanfranciscensis*	Ls-1001	NZ_RPFX00000000	Sourdough, Shanxi China	1,349,331	34.49	1386	(our lab; 20 May 2018)
*Fructilactobacillus sanfranciscensis*	Ah4	NZ_QFCR00000000.1	Sourdough, Anhui, China	1,368,476	34.68	1408	(our lab; 20 May 2018)
*Fructilactobacillus sanfranciscensis*	Gs2	NZ_QGEE00000000.1	Sourdough, Gansu China	1,373,332	34.53	1401	(our lab; 20 May 2018)
*Fructilactobacillus sanfranciscensis*	Gs9	NZ_QGEF00000000.1	Sourdough, Gansu China	1,365,822	34.43	1400	(our lab; 20 May 2018)
*Fructilactobacillus sanfranciscensis*	Ts9	GCA_006334515.1	Sourdough, Shanxi China	1,325,807	34.78	1386	(our lab; 20 May 2018)
*Fructilactobacillus sanfranciscensis*	Sd1_3	NZ_QGHM00000000.1	Sourdough, Shandong China	1,304,535	34.55	1335	(our lab; 20 May 2018)
*Fructilactobacillus sanfranciscensis*	Fs_1001	SAMN32652484	Sourdough, China	1,341,618	34.72	1367	(our lab; this study; 25 January 2023)
*Fructilactobacillus sanfranciscensis*	Fs_1002	SAMN32652485	Sourdough, China	1,318,495	34.61	1342	(our lab; this study; 25 January 2023)
*Fructilactobacillus sanfranciscensis*	Fs_1003	SAMN32652486	Sourdough, China	1,328,174	34.69	1368	(our lab; this study; 25 January 2023)
*Fructilactobacillus sanfranciscensis*	Fs_1004	SAMN32652487	Sourdough, China	1,333,883	34.75	1353	(our lab; this study; 25 January 2023)
*Fructilactobacillus sanfranciscensis*	Fs_1005	SAMN32652488	Sourdough, China	1,351,179	34.77	1374	(our lab; this study; 25 January 2023)
*Fructilactobacillus sanfranciscensis*	Fs_1006	SAMN32652489	Sourdough, China	1,299,074	34.57	1336	(our lab; this study; 25 January 2023)
*Fructilactobacillus sanfranciscensis*	Fs_1007	SAMN32652490	Sourdough, China	1,293,766	34.60	1315	(our lab; this study; 25 January 2023)
*Fructilactobacillus sanfranciscensis*	Fs_1008	SAMN32652491	Sourdough, China	1,305,974	34.70	1329	(our lab; this study; 25 January 2023)
*Fructilactobacillus sanfranciscensis*	Fs_1009	SAMN32652492	Sourdough, China	1,292,330	34.59	1315	(our lab; this study; 25 January 2023)
*Fructilactobacillus sanfranciscensis*	Fs_1010	SAMN32652493	Sourdough, China	1,302,254	34.70	1324	(our lab; this study; 25 January 2023)
*Fructilactobacillus sanfranciscensis*	Fs_1011	SAMN32652494	Sourdough, China	1,300,226	34.58	1336	(our lab; this study; 25 January 2023)
*Fructilactobacillus sanfranciscensis*	Fs_1012	SAMN32652495	Sourdough, China	1,302,328	34.59	1339	(our lab; this study; 25 January 2023)
*Fructilactobacillus sanfranciscensis*	Fs_1013	SAMN32652496	Sourdough, China	1,302,216	34.62	1338	(our lab; this study; 25 January 2023)
*Fructilactobacillus sanfranciscensis*	Fs_1014	SAMN32652497	Sourdough, China	1,300,972	34.58	1332	(our lab; this study; 25 January 2023)

**Table 2 microorganisms-12-00845-t002:** Identified EPS biosynthesis genes in the *Fructilactobacillus sanfranciscensis* genome.

Predicted Gene	NR Hit	NR Description
*epsA*	EKK20066.1	Cell envelope-associated transcriptional attenuator LytR-CpsA-Psr
	POH10471.1	LytR family transcriptional regulator
*epsB*	WP_041817972.1	CpsD/CapB family tyrosine-protein kinase
	AEN99639.1	Putative tyrosine-protein kinase capB
*epsD*	MVF15937.1	tyrosine-protein phosphatase
*wzx*	WP_103429181.1	bifunctional lysylphosphatidylglycerol flippase/synthetase MprF
	WP_046041031.1	flippase
	WP_014082292.1	flippase-like domain-containing protein
	WP_014081504.1	oligosaccharide flippase family protein
*wzy*	WP_139571209.1	polysaccharide polymerase
	WP_198988114.1	O-antigen polysaccharide polymerase Wzy
*rgpI, waaB*	WP_139571129.1	Glycosyltransferase

**Table 3 microorganisms-12-00845-t003:** Antibiotic resistance of 14 strains of *Fructilactobacillus sanfranciscensis*.

	Antibiotics	Aminoglycolsides			Sulfa-Mido	Tetra- Cycline	Penicillins		Chloram-Phenicol	Linco- Samides	Macro Lide
Strains		CN	K	S	SXT	TE	OXO	AMP	C	CD	E
Fs_1001		R	R	R	R	IR	IR	S	S	S	S
Fs_1002		R	R	R	R	IR	S	S	S	S	S
Fs_1003		R	R	R	R	R	S	S	S	S	S
Fs_1004		R	R	R	R	IR	IR	S	S	S	S
Fs_1005		R	R	R	R	R	S	S	S	S	S
Fs_1006		R	R	R	R	R	S	S	S	S	S
Fs_1007		R	R	R	R	R	R	S	S	S	S
Fs_1008		R	R	R	R	R	IR	S	S	S	S
Fs_1009		R	R	R	R	R	S	S	S	S	S
Fs_1010		R	R	R	R	IR	S	S	S	S	S
Fs_1011		R	R	R	R	R	S	S	S	S	S
Fs_1012		R	R	R	R	R	S	S	S	S	S
Fs_1013		R	R	R	R	IR	IR	S	S	S	S
Fs_1014		R	R	R	R	R	IR	S	S	S	S

Gentamicin, 10 μg (CN); Kanamycin, 30 μg (K); Streptomycin, 10 μg (S); Erythromycin, 15 μg (E); Clindamycin, 2 μg (CD); Oxacilin, 1 μg (OXO); Ampicillin, 10 μg (AMP); Tetracycline, 30 μg (TE); Chloramphenicol, 30 μg (C); Trimethoprim-sulfamethoxazole, 25 μg (SXT). The antibiotic resistance of the 14 *F. sanfranciscensis* strains were classified as susceptible (S, zone diameter > 20), intermediate (IR, 15 < zone diameter < 19), or resistant (R, zone diameter ≤ 14).

## Data Availability

The genome sequences are available in the NCBI, and Table 1 has listed all strains’ accession numbers.

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
