# Peer review of "Comparative Genomics Reveals Genetic Diversity and Variation in Metabolic Traits in Fructilactobacillus sanfranciscensis Strains"

_microorganisms, 2024, doi:10.3390/microorganisms12050845_

Round 1
Reviewer 1 Report
Comments and Suggestions for Authors
Suggestions for improving the manuscript have been given in the attached pdf file.

Reviewer 2 Report
Comments and Suggestions for Authors
“D-glucose, D-fructose, D-maltose, D-trehalose, D-sucrose, D-lactose, D-galactose, L-
arabinose, soluble starch, and mannitol were used as sole carbon sources to verify the
expression of carbohydrate utilization genes in 14 F. sanfranciscensis strains. Ten different carbohydrates were added individually into mMRS medium at the concentration of 2 g/100 mL followed by the addition of 0.5% (w/v) bromocresol purple solution at the ratio of 1.5% (v/v)” –
In order to determine the fermentation profile, the API CHL test should have been used, then a full panel of the fermentation profile would be obtained. In order to determine the fermentation profile, the API CHL test should have been used, then a full panel of the fermentation profile would be obtained. If possible, I recommend adding this experiment.
Why wasn't a vancomycin resistance test performed? When testing lactic acid bacteria, confirming vancomycin resistance is very important because it is an innate activity and its absence may indicate an unusual antibiotic resistance profile for this group of microorganisms.
I have no further comments. The work is written carefully and the methods used are appropriate.
Author Response
Please see the attachment. Please see the latest document for our response to comments.
